# FREQUENCY-PRIOR GUIDED DIFFUSION FOR NIGHT-TIME FLARE REMOVAL

## ABSTRACT

Nighttime photography with intense light sources frequently produces significant flare artefacts that obscure the background, resulting in diminished image quality. Existing encoder–decoder methods can remove flare, but when trained on limited datasets, they still suffer from some issues: residual artifacts and color shifts. In contrast, diffusion-based methods can address these problems to some extent, but the multi-step diffusion process leads to error accumulation, which in turn causes background distortion. To address the above issues, we propose a novel single-step diffusion framework, **FGDNet**, for nighttime flare removal, guided by Laplace Pyramid frequency priors. Specifically, Our method leverages stable diffusion combined with frequency prior guidance to achieve high-fidelity flare removal without requiring flare annotations. The framework consists of three key components: (1) A **L**atent **D**iffusion-based **D**eflare **M**odule (LDDM) that performs flare removal and preliminary background reconstruction through single-step diffusion with LoRA fine-tuning; (2) A **M**ulti-scale **F**requency **I**njection **M**odule (MFIM) that extracts high-frequency details through Laplacian pyramid decomposition, aligns authentic textures, and injects them into the VAE decoder to restore fine details; (3) A **M**ulti-band **F**requency **F**usion **M**odule (MFFM) that employs multi-reference attention to adaptively fuse preliminary results with high/low-frequency information from the input image, further enhancing structural and color restoration. Experiments on Flare7K and Flare7K++ show superior performance in PSNR, SSIM, LPIPS, and no-reference metrics (MUSIQ, MANIQA), reducing artifacts while enhancing background detail and color fidelity in complex nighttime scenes.

## 1 INTRODUCTION

Lens flare refers to irregular bright spots in images caused by refraction and scattering within the lens system when using smartphones to capture nighttime scenes with light sources, which often degrades image quality and aesthetics. Existing flare removal methods can be categorized into encoder–decoder-based approaches and diffusion-based methods.

More specifically, encoder–decoder approaches (Dai et al., 2022; 2024; Song & Bae, 2023; Zhou et al., 2023; Chen et al., 2024; Kotp & Torki, 2024; Deng et al., 2024; Ma et al., 2025) typically focus on training general image restoration models or carefully designing network architectures to learn flare features for removal. (Dai et al., 2022; 2024) constructed datasets for nighttime flare removal based on the physics of flare formation, paving the way for supervised training. Building on these datasets, (Zhang et al., 2023a) built a Swin-Transformer model with Spatial Frequency Block and Fast Fourier Convolution for long-range and global frequency features. (Zhou et al., 2023) revised ISP auto-exposure to avoid global brightness shifts and local saturation in synthetic data. (Chen et al., 2024) introduced LPFSformer with location priors to better localize and suppress flare. (Kotp & Torki, 2024) integrated depth estimation into Uformer for flare removal. (Deng et al., 2024) proposed a knowledge-driven flare localization approach that predicts the degree of degradation by leveraging gradient preservation and template matching rules. (Ma et al., 2025) proposed SGSFT, a self-prior guided spatial-frequency transformer for nighttime flare removal. While these methods can remove flares, their performance in real nighttime scenes is limited by the training datasets, resulting in residual artifacts and background color shifts. Diffusion-based methods can achieve high-quality visual results, as they are trained on vast datasets with rich prior knowledge,

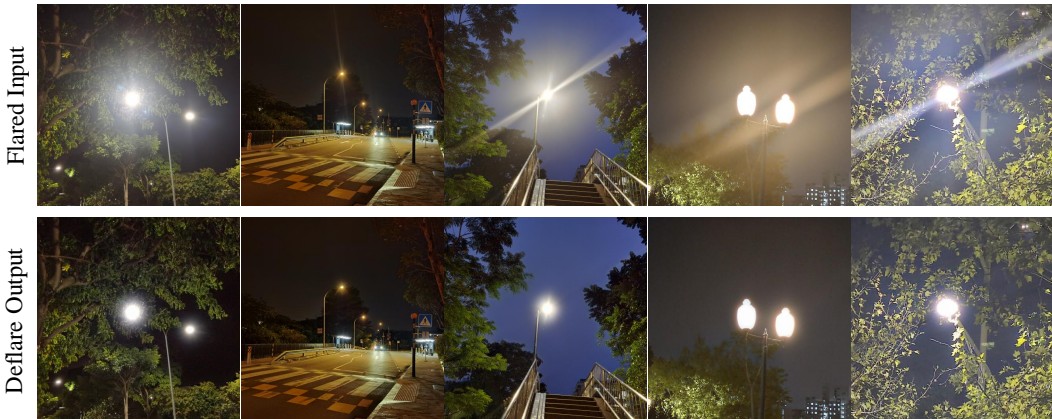

Figure 1: Deflaring results on complex flare-corrupted images by our method. By integrating stable diffusion and laplacian pyramid frequency prior, our proposed FGDNet effectively removes flares, preserves image color and structure, and produces high-quality results.

thereby opening up possibilities for flare removal (Zhou et al., 2024). (Zhou et al., 2024) proposed a multi-step diffusion approach for flare removal that operates in latent space by leveraging generative priors from pre-trained diffusion models, while preserving image structure through Structural Guidance Injection and Adaptive Feature Fusion that incorporates a luminance gradient prior. (Wang et al., 2024) processed images based on latent diffusion models, employing time-aware encoders for conditional control and controllable feature encapsulation to achieve image structural fidelity. (Lin et al., 2024) decomposed the task into degradation removal and content regeneration, utilizes latent diffusion for reconstruction, and introduces IRControlNet with region-adaptive guidance to balance naturalness and fidelity. However, the aforementioned methods employ conditional control/VAE decoding to fuse original image structure information, often depending on damaged source data. Moreover, the multi-step diffusion process accumulates inference errors, resulting in background structure loss.

To address these problems, we focus on proposing a more effective method based on latent diffusion models that not only resolves flare artifact residual and color shift issues but also achieves efficient inference and preserves image structure. Inspired by the reversible and closed-form band decomposition framework of the Laplace pyramid (LP) (Burt & Adelson, 1987), illumination or color is mainly expressed in the low-frequency components, and content details are more related to the high-frequency components. Therefore, we propose **FGDNet**, a frequency-guided one-step diffusion framework for nighttime flare removal, where frequency components are obtained via LP decomposition. FGDNet comprises three key components: a **L**atent **D**iffusion-based **D**eflare **M**odule (LDDM), which uses LoRA fine-tuning and one-step diffusion to leverage Stable Diffusion's rich priors and produce a latent representation of the deflared result; a **M**ulti-scale **F**requency **I**njection **M**odule (MFIM), which decomposes the flare-degraded image into multi-scale frequency components via LP, aligns these frequencies with ground truth at the texture level, and injects them into the VAE decoder to enhance LDDM's generative capacity and yield initial deflare results; and a **M**ulti-band **F**requency **F**usion **M**odule (MFFM), which employs a multi-reference attention mechanism to fuse the initial deflared output with the corresponding frequency bands of the original image while jointly referencing other bands from both sources, followed by image reconstruction to obtain a visually enhanced result. The visual effects are shown in Fig .1

Our main contributions are summarized as follows:

(1) We propose a novel single-step diffusion framework for nighttime flare removal, guided by Laplace Pyramid frequency priors.

(2) We construct a **M**ulti-scale **F**requency **I**njection **M**odule (MFIM) to align high-frequency details across multiple scales in the Laplacian pyramid of the input image and GT, then inject these details into the VAE decoder to mitigate background distortion.

(3) We construct a **M**ulti-band **F**requency **F**usion **M**odule (MFFM) to further enhance visual fidelity by integrating different information across frequency bands from the input and initial deflare results.

## 2 RELATED WORK

**Flare Removal:** Early methods employed hardware optimizations (Raskar et al., 2008; Boynton & Kelley, 2003; Macleod & Macleod, 2010) or two-stage detection-removal pipelines (Asha et al., 2019; Chabert, 2015; Vitoria & Ballester, 2019), but faced cost limitations and poor generalization to complex scenarios. With the growing popularity of deep learning methods, several studies (Qiao et al., 2021; Jin et al., 2022; Wu et al., 2024c; He et al., 2025) have achieved flare removal without labeled training datasets, but with unstable performance. With the growing popularity of encoder-decoder models, numerous encoder-decoder-based methods have achieved flare removal using the lens flare removal datasets proposed by (Wu et al., 2021; Dai et al., 2022; 2024). Zhang et al. (2023a) incorporated Spatial Frequency Block and Fast Fourier Convolution into a Swin-Transformer to capture global frequency features. Song & Bae (2023) designed a cascade network with comparative learning on tripled outputs. Zhou et al. (2023) revised the ISP automatic exposure principle to improve synthetic data quality. Chen et al. (2024) introduced LPFSformer for flare localization and suppression via prior injection. (Deng et al., 2024) propose a knowledge-driven flare localization approach that predicts the degree of degradation by leveraging gradient preservation and template matching rules. Kotp & Torki (2024) achieves flare removal by injecting depth estimates from flare-damaged images into the Uformer model. Ma et al. (2025) proposed a self-prior guided spatial-frequency transformer. However, the appeal method is constrained by limited datasets and exhibits poor practical generalisability.

**Latent Diffusion based Image Restoration:** With the growing popularity of pre-trained diffusion models, leveraging their powerful generative capabilities while preserving image structural details has become a key research focus in the field of flare removal. (Zhou et al., 2024) proposes a multi-step diffusion approach for flare removal that operates in latent space by leveraging generative priors from pre-trained diffusion models, while preserving image structure through structural guidance injection and adaptive feature fusion that incorporates a luminance gradient prior. In other fields utilising potential diffusion models for image processing, (Wang et al., 2024) processes images based on latent diffusion models, employing time-aware encoders for conditional control and controllable feature encapsulation to achieve image structural fidelity. (Lin et al., 2024) decomposes the task into degradation removal and content regeneration, utilizes latent diffusion for reconstruction, and introduces IRControlNet with region-adaptive guidance to balance naturalness and fidelity. (Jiang et al., 2024) proposes an image restoration framework that automatically identifies and repairs diverse unknown degradations through blind quality assessment and structural-corrected latent diffusion. However, using the original image to condition the UNet or incorporating it into the VAE encoder relies on the damaged information in the source image, while multi-step diffusion accumulates errors that lead to distortion in the background structure.

**Laplacian Pyramid:** Image Laplace Pyramid (LP) decomposition (Burt & Adelson, 1987) is a multi-scale representation technique that decomposes an image into different resolution layers, each capturing a specific frequency band. This structure consists of a band-pass-filtered image with each level representing the difference between neighboring resolution versions of the original image. Existing work on CNN-based image processing applies the difference structure of the LP paradigm (Denton et al., 2015; Ghiasi & Fowlkes, 2016; Lai et al., 2017; Liang et al., 2021; Luo et al., 2023; Zhang et al., 2023b; Atzmon et al., 2024). (Lai et al., 2017) employs a Laplacian Pyramid Super-Resolution Network (LapSRN) that progressively reconstructs high-resolution images by predicting high-frequency residuals at each pyramid level and upsampling to finer levels through a single forward pass. (Liang et al., 2021)proposes a Laplace pyramid-based image translation network for real-time 4K image translation by processing low-frequency attribute transforms and high-frequency details separately. (Luo et al., 2023) utilizes a Laplacian pyramid structure to decompose multi-modal images into different resolution levels, processing high-frequency components with CNN and low-frequency components with transformers across pyramid levels for effective image fusion. (Zhang et al., 2023b) employs Laplace pyramid decomposition for HDR-to-LDR conversion, using adaptive 3D LUT for low frequencies and learned Laplace filters for high frequencies to achieve both global tone mapping and local detail preservation. (Atzmon et al., 2024) generates multi-resolution images by Laplace pyramid-based noising with varied frequency attenuation and progressive denoising.

**Differently**, we proposed FGDNet, a single-step diffusion framework based on Laplacian pyramid frequency priors for nighttime flare removal. Our method integrates stable diffusion models with frequency prior guidance to slove residual flare, color shifts and structural distortion.

## 3 METHOD

Fig. 2 illustrates the overall pipeline of our FGDNet. The flare-degraded image $I_F$ is processed by a U-Net within the LDDM to generate a deflare latent representation $\hat{z}_0$. The high-frequency layers $\{h_1^{(f)}, h_2^{(f)}, h_3^{(f)}\}$ obtained from Laplace Pyramid decomposition $(LP(\cdot))$ of $I_F$ are structurally aligned with the GT via the MFIM, producing refined frequency layers $\{h_1^{(m)}, h_2^{(m)}, h_3^{(m)}\}$. These frequency layers are transformed by a feature mapping layer and then injected into the VAE decoder of the LDDM. LDDM produces preliminary deflare results $\hat{I}'_{out}$ by single-step diffusion. The MFFM fuses the multi-band sequences $\{h_1^{(f)}, h_2^{(f)}, L^{(f)}\}$ and $\{h^{(d)}1, h^{(d)}2, L^{(d)}\}$ from $LP(I_F)$ and $LP(\hat{I}_{out})$ into $\{h_1^{(o)}, h_2^{(o)}, L^{(o)}\}$, ultimately reconstructing the high-quality output $\hat{I}_{out}$.

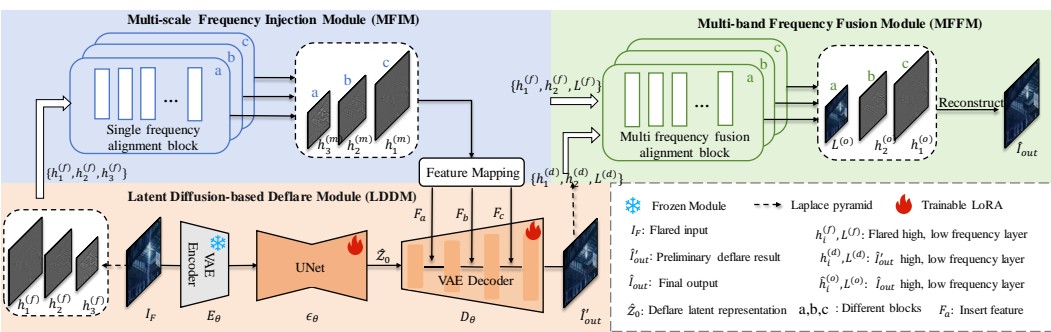

Figure 2: Our FGDNet training pipeline consists of three key components: Latent Diffusion-based Deflare Module (LDDM), Multi-scale Frequency Injection Module (MFIM), and Multi-band Frequency Fusion Module (MFFM).

### 3.1 DEFLARE VIA LATENT DIFFUSION MODEL

Our Latent Diffusion-based Deflare Module (LDDM) is composed of a UNet $\epsilon_\theta$, a pre-trained VAE encoder $E_\theta$, and a VAE decoder $D_\theta$. Given a flare-damaged image $I_F$, we first obtain its latent representation via the encoder, $z_l = E_\theta(I_F)$, and then extract a text prompt using a prompt extractor (Wu et al., 2024b), $c_p = \mathcal{D}(I_F)$. Conventional diffusion models (Wang et al., 2024; Lin et al., 2024), employing a multi-step diffusion process, diffuse the latent $z_l$ over time steps $t \in [1, \ldots, T]$ to produce $z_t = \alpha_t z_l + \beta_t$ (with time-dependent $\alpha_t$ and $\beta_t$) and then predict noise $\hat{\epsilon}$ with UNet to recover $\hat{z}_0 = \frac{z_t - \beta_t \hat{\epsilon}}{\alpha_t}$. However, these methods suffer from texture distortion due to iterative noise. The single-step diffusion method reduces such distortion and errors by completing the mapping in a single pass. Therefore, we employ a single-step diffusion method (Wu et al., 2024a) to denoise the latent variable $z_l$ by $\epsilon_\theta$ during the inference process. Conditioned on the text prompt $c_l$, the denoising process is formulated as $\hat{z}_0 = \epsilon\theta(z_l, t_f, c_p)$. $t_f$ denotes fixed time step, $\hat{z}_0$ is the denoised latent representation, which is subsequently converted into a pixel representation via $D_\theta$.

To fine-tune diffusion models, the primary approaches are full fine-tuning and parameter-efficient fine-tuning (PEFT). Given the high computational cost of the former, existing tasks mainly employ PEFT methods. We leverage Low-Rank Adaptation (LoRA) (Hu et al., 2022), an efficient fine-tuning method that achieves effective parameter adjustment through low-rank decomposition. Given a pre-trained original weight matrix $W_0$, the update process is formulated as: $W = W_0 + \Delta W$. During training, $W_0$ remains frozen, while low-rank increment $\Delta W$ is updated.

During training, we observed that the VAE decoder focuses more on structural image reconstruction compared to the UNet. Therefore, directly injecting texture information as a conditioning signal into the decoding process via our MFIM proves more effective. To enable efficient fine-tuning of the diffusion model and adapt to MFIM's injection mechanism, we apply LoRA to both the UNet and the VAE decoder. The fine-tuned UNet generates deflare-optimized latent representations, while the LoRA-adapted decoder reconstructs high-quality images with enhanced detail through our proposed injection mechanism.

(a) Laplace pyramid decomposition visualization      (b) Multi-reference attention mechanism

Figure 3: Detail of Laplace pyramid frequency layers and multi-reference attention mechanism.

## 3.2 FREQUENCY PRIOR STRATEGY BASED LAPLACIAN PYRAMID

The Laplacian Pyramid (LP) decomposition technique (Burt & Adelson, 1987) separates an image into distinct frequency bands via a multi-scale representation. Owing to its effectiveness in revealing textural and structural information within images, this decomposition method has demonstrated significant value in various image processing tasks (Liang et al., 2021; Lai et al., 2017). Building on this foundation, we observe through the LP framework that illumination and color information are primarily distributed in low-frequency components, while structural and textural details predominantly manifest in high-frequency components, a decomposition highly consistent with the inherent structural hierarchy of images.

However, flare degradation presents a more complex scenario, as it is not confined to a single frequency band. As a high-intensity localized artifact, flare manifests not only in high-frequency components (e.g., edges and textures) but also significantly affects low-frequency layers that primarily encode brightness and color information (See in Fig. 3 (a)). To address this multi-frequency nature of flare artifacts, we introduce a frequency prior strategy based on LP decomposition. Specifically, given a flare-damaged image $I_F$, we obtain different frequency components $\mathbf{L}_n(I_F) \in \mathbb{R}^{\frac{H}{2^{n-1}} \times \frac{W}{2^{n-1}}}$ via the LP decomposition operation:

$$\mathbf{L}_k = G_k(I) - \delta_{k<n} \cdot U_p(G_{k+1}(I)), \tag{1}$$

in which $G_k$ and $U_p$ denote the standard Gaussian pyramid and its upsampling operation, respectively. In the constructed Laplacian pyramid $\mathbf{L}_n(I_F)$, the lower $n-1$ levels capture high-resolution images/features rich in detail and texture, while the top layer contains low-frequency image content with the lowest spatial resolution. Therefore, we introduced a frequency prior strategy based on LP decomposition: high-frequency components serve as priors for reconstructing textural details during the decoding process of the diffusion model, while the integration of high- and low-frequency components provides priors for enhancing structure consistency and restoring color fidelity.

Based on the frequency prior strategy, we designed two core modules:

(1) The Multi-scale Frequency Injection Module (MFIM) injects multi-scale high-frequency components into the VAE decoder after structurally aligning them with the ground truth (GT) image, thereby ensuring accurate detail restoration during the image reconstruction process in the LDDM. (2) The Multi-band Frequency Fusion Module (MFFM) integrates the initial deflared output with corresponding frequency information from the input image using a multi-reference attention mechanism. By multi-band fusion, it enhances structural and color fidelity to produce refinements.

## 3.3 STRUCTURE RESTORATION VIA FREQUENCY PRIOR

In MFIM, we set $n = 4$ in equation 1 and extract the high-frequency component sequence $\{h_1^{(f)}, h_2^{(f)}, h_3^{(f)}\}$ as strong prior conditions. For the high-frequency components at each scale $(h_i^{(f)} \in \mathbb{R}^{\frac{H}{2^{i-1}} \times \frac{W}{2^{i-1}} \times C})$, we employ a single frequency alignment block composed of multiple self-attention transformer blocks to process $h_i^{(f)}$. The self-attention transformer block performs modeling based on multi-head self-attention mechanism (SA), with SA formula defined as follows:

$$\mathrm{SA}(\mathbf{Q}, \mathbf{K}, \mathbf{V}) = \mathrm{Softmax}\left(\frac{\mathbf{Q}\mathbf{K}^\top}{\sqrt{d_k}}\right)\mathbf{V}, \tag{2}$$

where $\mathbf{Q}, \mathbf{K}, \mathbf{V}$ are the query, key and value matrices; $d$ is the dimension of the query and key vectors. Finally, the high-frequency bands $\{h_1^{(m)}, h_2^{(m)}, h_3^{(m)}\}$ obtained through multiple single frequency alignment blocks are fed into a feature mapping module. The feature mapping module performs a series of progressive convolutional operations that gradually increase the channel dimension while reducing spatial resolution, constructing multi-scale deep semantic features $\{F_a, F_b, F_c\}$ to support feature fusion and reconstruction in the upsampling path of the VAE decoder. Following the reconstruction process through the VAE decoder, a preliminary result $\hat{I}'_{out}$ can be obtained.

In MFFM, we set $n = 3$ in equation 1 to obtain the frequency pyramid for the input image $I_F$ and the preliminary deflare result $\hat{I}'_{out}$ using LP decomposition. This process yields the frequency bands $\{h_1^{(f)}, h_2^{(f)}, L^{(f)}\}$ from $I_F$ and $\{h_1^{(d)}, h_2^{(d)}, L^{(d)}\}$ from $\hat{I}'_{out}$. For each pair $p$ of the frequency component combination list $\mathcal{P} = \{(h_1^{(d)}, h_1^{(f)}), (h_2^{(d)}, h_2^{(f)}), (L^{(d)}, L^{(f)})\}$, we employ a multi frequency fusion alignment block composed of multiple multi-reference attention transformer blocks. The multi-reference attention transformer block based on multi-reference attention mechanism (MA). Integrating frequency combinations $p$, information from other combinations within $\mathcal{P}$ is referenced.

Within MA (See in Fig. 3 (b)), one feature is designated as the primary feature $\mathbf{x}_m$, while the others serve as reference features $\mathbf{x}_{r_1}, \mathbf{x}_{r_2}$. First, the primary feature undergoes a linear projection layer to generate the query $\mathbf{Q}_m$, key $\mathbf{K}_m$, and value $\mathbf{V}_m$. Reference features were projected to generate keys $\mathbf{K}_{r_1}, \mathbf{K}_{r_2}$ and values $\mathbf{V}_{r_1}, \mathbf{V}_{r_2}$. The MA is implemented as follows:

$$\text{MA}(\mathbf{Q}_m, \mathbf{K}_i, \mathbf{V}_i) = \text{Softmax}\left(\frac{\mathbf{Q}_m \mathbf{K}_i^\top}{\sqrt{d_k}}\right) \mathbf{V}_i, \quad i \in (m, r_1, r_2) \tag{3}$$

This attention mechanism computes matrix multiplication between the main feature query $\mathbf{Q}_m$ and $\mathbf{K}_m, \mathbf{K}_{r_1}, \mathbf{K}_{r_2}$ respectively. Following softmax normalization, it yields attention scores $A_m, A_{r_1}, A_{r_2}$. These are then combined with $\mathbf{V}_m, \mathbf{V}_{r_1}, \mathbf{V}_{r_2}$ through matrix multiplication to produce the final output $\{\mathbf{x}'_m, \mathbf{x}'_{r_1}, \mathbf{x}'_{r_2}\}$. We concatenate the outputs along the channel dimension and pass them through a fusion projection layer to perform information fusion, yielding the corresponding fused features $\mathbf{x}'_m$.

Finally, we employ the fused frequency pyramid $\{h_1^{(o)}, h_2^{(o)}, L^{(o)}\}$ to reconstruct the original image by iteratively descending through the pyramid hierarchy. This process involves upsampling and superimposing high-frequency details layer by layer, ultimately yielding precise colour output $\hat{I}_{out}$.

## 3.4 TRAINING STRATEGY

Unlike previous multi-stage training approaches, Our FGDNet is an end-to-end framework that is trained in a single stage. We utilize the data consistency loss $\mathcal{L}_{data}$ to train LDDM, evaluating the difference between the preliminary deflare result $\hat{I}'_{out}$ and the ground truth (GT). The frequency consistency loss $\mathcal{L}_{freq}$ is employed to train MFIM, assessing the discrepancy between $\{h_1^{(m)}, h_2^{(m)}, h_3^{(m)}\}$ and the high-frequency pyramid of GT. We jointly train the MFFM using $\mathcal{L}_{data}$ and $\mathcal{L}_{freq}$, evaluating the difference between the final result $\hat{I}_{out}$ and GT. In summary, we assess the discrepancy between model results and GT across both pixel and frequency dimensions.

**Data consistency loss:** Our data consistency loss is composed of the mean squared error loss $\mathcal{L}_{MSE}$ and the perceptual similarity loss $\mathcal{L}_{LPIPS}$, as expressed by the following formula:

$$\mathcal{L}_{MSE} = \|S - T\|_2, \quad \mathcal{L}_{LPIPS} = \sum_l \|w^l \odot (\hat{S}^l - \hat{T}^l)\|_2^2, \tag{4}$$

$$\mathcal{L}_{data} = \lambda_1 \mathcal{L}_{MSE}(S, T) + \lambda_2 \mathcal{L}_{LPIPS}(S, T). \tag{5}$$

in which $S$ is the module's output image, $T$ is the GT, $w_l$ is a learnable weight, $\hat{S}^l$ and $\hat{T}^l$ are the channel-wise unit-normalized feature maps from the $l$-th layer of VGG network for $S$ and $T$.

**Frequency consistency loss:** Our frequency consistency loss employs mean squared error loss to constrain different frequency layers, as expressed by the following formula:

$$\mathcal{L}_{freq}(S, T) = \sum_j 2^j \|\mathbf{L}_j(S) - \mathbf{L}_j(T)\|_2. \tag{6}$$

where $j \in [1, 2, \cdots n]$ and $\mathbf{L}_j(S)$ is the $j$-th level of the Laplacian pyramid representation of $S$. $2^j$ is the weighting coefficient for the j-th layer.

# 4 EXPERIMENTS

## 4.1 EXPERIMENTAL DETAILS

**Datasets:** Our method is trained on the Flare7K (Dai et al., 2022) and Flare7K++ (Dai et al., 2024) datasets. Flare7K contains 5,000 scattered and 2,000 reflected flare images. The synthesis pipeline applies inverse gamma correction ($\gamma \sim U(1.8, 2.2)$), random RGB scaling ($U(0.5, 1.2)$), and Gaussian noise ($\sigma^2 \sim 0.01\chi^2$) to flare-free images from 24K Flickr (Zhang et al., 2018b), then blends them with randomly transformed (rotation, translation, scaling, etc.) flare components to generate degraded images. Flare7K++ extends Flare7K by adding 962 real flare-only images (Flare-R) captured with various camera lenses. We mix Flare7K and Flare-R images (Dai et al., 2024) 1:1 to create the Flare7K++ dataset. Our experiments follow the synthesis settings of Flare7K/Flare7K++ and are evaluated on the Flare7K++ test set containing 100 real sample pairs.

**Experimental settings:** Our FGDNet employs the Adam optimizer during training, with learning rate, total iterations, and batch size set to 5e-5, 180k, and 2 respectively. The LoRA level in the UNet and VAE decoder is set to 4. The SD 2.1-base is used as the pre-trained T2I model. The weighted scalars $\lambda_1$ and $\lambda_2$ are set to 2 and 1, respectively. During evaluation, we employ reference metrics PSNR, SSIM (Wang et al., 2004), and LPIPS (Zhang et al., 2018a), along with non-reference metrics MANIQA (Yang et al., 2022) and MUSIQ (Ke et al., 2021) as target objectives.

**Comparison approaches:** Based on the Flare7K training dataset, we compared our method with existing flare removal methods (Dai et al., 2022; Zhou et al., 2023) for comparison. These methods were trained without light source labels. We also compared previous work trained on the Daytime flare dataset (Wu et al., 2021). For light source restoration after flare removal, we employed the same brightness-threshold segmentation approach as (Dai et al., 2022). Based on the Flare7K++ training dataset, we compared our method with existing flare removal methods (Dai++ (Dai et al., 2024), Kotp (Kotp & Torki, 2024), Ma (Ma et al., 2025)) trained using light source and flare labels. Notably, our method does not utilize flare labels for training, whereas the aforementioned approaches all employ flare labels.

## 4.2 EXPERIMENTAL RESULTS

**Qualitative Comparison:** On real-world nighttime flare datasets, we conducted a series of qualitative comparisons with state-of-the-art flare removal methods. Fig. 4 displays the visual results of existing methods on the real-world paired flare dataset provided by (Dai et al., 2024), while Fig. 5 shows their performance on the real-world captured dataset from the same study. The visual results in Fig. 4 demonstrate that our method effectively removes complex flare combinations and large-area flares without introducing artifacts, while successfully restoring the colors of background regions obscured by flares, achieving superior visual quality compared to other methods. The results in Fig. 5 indicate that existing methods exhibit limited generalization in real nighttime flare scenes, often leaving color artifacts after flare removal. In contrast, our method outperforms others in visual performance (rows 1) and also shows effective removal of multi-colored flares (rows 2–3).

**Quantitative Comparison:** Our FGDNet achieves strong performance on the real-world nighttime flare dataset provided by (Dai et al., 2024). Results are shown in Table 1. Among methods trained on the Flare7K dataset, our approach (Ours1) achieves the best performance on objective evaluation metrics. For reference-based metrics PSNR, SSIM, and LPIPS, our results surpass the second-place method by 0.57, 0.01, and 0.006, respectively. For non-reference metrics MUSIQ and MANIQA, our results outperform the second-place method by 0.28 and 0.0048, respectively. For methods trained on the Flare7K++ dataset, our approach (Ours) achieved higher PSNR and SSIM scores than the second-best method by 0.19 and 0.001, respectively. For MUSIQ and MANIQA metrics, our method outperformed the second-best approach by 0.11 and 0.0038, respectively.

## 4.3 ABLATION STUDY

**Effectiveness of LDDM:** We removed the MFIM and MFFM from FGDNet to validate the effectiveness of the LDDM module. As shown in Fig. 6 (a) (LDDM), our LDDM module effectively removes nighttime flares, with the text structure in the lower-left corner exhibiting minimal distortion, highlighting the advantage of the single-step diffusion approach in texture preservation. LDDM

Table 1: Quantitative comparison on the Flare7K++ real test dataset. ↓/↑ denote lower/higher is better. Best and second-best results are in black and underlined, respectively.

| Metric\Method | Previous work | Works trained on Flare7K | | | Works trained on Flare7K++ | | | |
| | Wu | Dai | Zhou | Ours1 | Dai++ | Kotp | Ma | Ours |
|---|---|---|---|---|---|---|---|---|
| PSNR↑ | 25.15 | 26.60 | 25.18 | **27.17** | 27.63 | 27.66 | 28.08 | **28.27** |
| SSIM↑ | 0.883 | 0.892 | 0.872 | **0.902** | 0.894 | 0.897 | 0.904 | **0.905** |
| LPIPS↓ | 0.0576 | 0.0511 | 0.0548 | **0.0451** | 0.0428 | 0.0422 | **0.0417** | 0.0432 |
| MUSIQ↑ | 59.27 | 59.09 | 59.09 | **59.55** | 59.29 | 59.05 | 59.60 | **59.71** |
| MANIQA↑ | 0.6257 | 0.6262 | 0.6304 | **0.6352** | 0.6283 | 0.6282 | 0.6377 | **0.6415** |

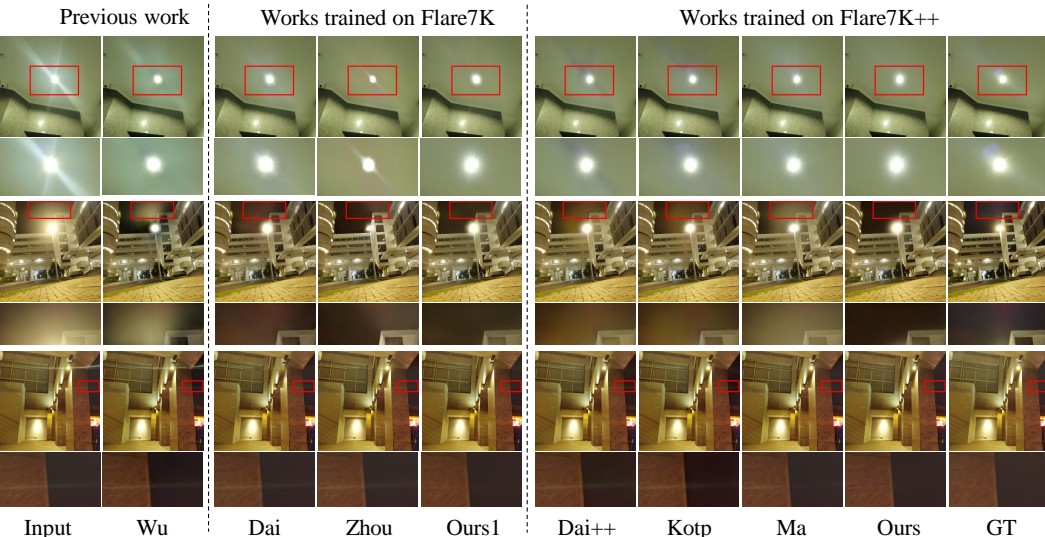

Figure 4: Qualitative comparison on the real-world paired test dataset (with GT) provided by (Dai et al., 2024).

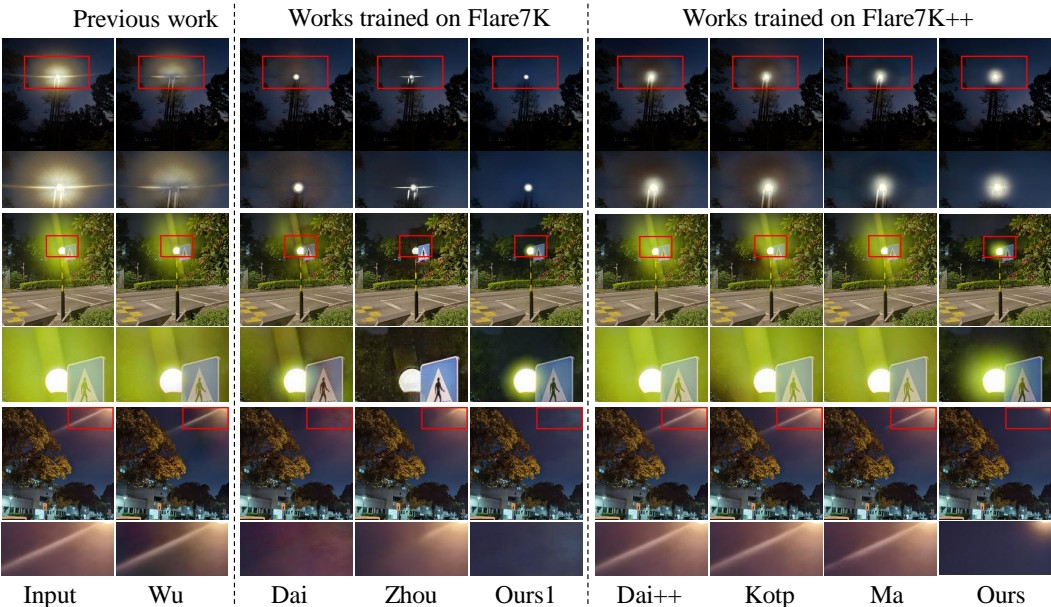

Figure 5: Qualitative comparison on the real-world captured dataset (without GT) provided by (Dai et al., 2024).

demonstrates limited performance in color restoration. By calculating the colour map (Wang et al., 2021), it can be observed that the LDDM results have color shifts (red box) compared to the GT.

**Effectiveness of MFIM:** Building upon the LDDM, we integrated the MFIM module to validate the additional improvement it provides. While our LDDM module successfully removes flares, structural discrepancies persist. To address this, we proposed the MFIM, which injects image high-frequency layers as supplementary information into the VAE decoder to assist reconstruction. As shown in Fig. 6(a) (LDDM+MFIM), the text structure in the lower-left corner aligns well with the GT. Quantitative results in Table 3 demonstrate that incorporating the MFIM significantly improves performance metrics. Our MFIM effectively preserves background texture details while avoiding the introduction of high-frequency artefacts caused by flare, significantly enhancing night-time flare suppression and background recovery performance.

Table 2: Ablation Study of FGDNet

| LDDM | MFIM | MFFM | PSNR↑ | SSIM↑ | LPIPS↓ |
|------|------|------|-------|-------|--------|
| ✓ |  |  | 25.05 | 0.811 | 0.0687 |
| ✓ | ✓ |  | 26.84 | 0.899 | 0.0461 |
| ✓ |  | ✓ | 27.09 | 0.898 | 0.0447 |
| ✓ | ✓ | ✓ | **27.17** | **0.902** | 0.0447 |

Table 3: Ablation study of MA

W/O reference: $SA(\mathbf{x}_m) \rightarrow \mathbf{x}'_m$
W/ reference: $MA(\mathbf{x}_m, \mathbf{x}_{r1}, \mathbf{x}_{r2}) \rightarrow \mathbf{x}'_m$

| Method | PSNR↑ | SSIM↑ | LPIPS↓ |
|--------|-------|-------|--------|
| W/O reference | 26.83 | 0.89 | 0.0484 |
| W/ reference | **27.09** | **0.898** | **0.0447** |

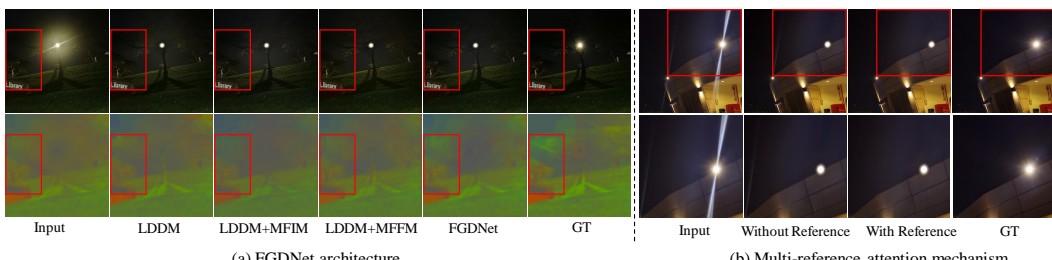

Input    LDDM    LDDM+MFIM    LDDM+MFFM    FGDNet    GT      Input   Without Reference   With Reference   GT

(a) FGDNet architecture        (b) Multi-reference attention mechanism

Figure 6: Visual results from ablation studies on our method. (a) Results of the FGDNet architecture. (b) Results of the multi-reference attention mechanism in our MFFM.

**Effectiveness of MFFM:** Based on the LDDM, we incorporated the MFFM to validate its effectiveness. The red-boxed region in Fig. 6(a) (LDDM+MFFM) demonstrates the module's ability to preserve texture details, while its color map indicates improved color retention. Subsequently, integrating the MFFM with the LDDM and MFIM forms the complete FGDNet. The color map in the red-boxed region of Fig. 6(a) (FGDNet) shows that the resulting color characteristics are closer to the GT. As shown in Table 3, the inclusion of the MFFM enhances the model's performance in both flare removal and color retention. We performed ablation experiments on the multiple reference attention mechanism to validate its efficacy. We primarily validated two schemes: one incorporates references to other features ($\mathbf{x}_{r1}$, $\mathbf{x}_{r2}$) when processing the $\mathbf{x}_m$ feature, while the other employs self-attention without external references. Both visually (Fig. 6) and quantitatively (Table 3), the multi-reference attention mechanism outperforms the non-reference mechanism, validating that incorporating reference frequency layers enhances restoration performance.

# 5 CONCLUSION

This paper addresses the issues of residual flare, color distortion, and background structure distortion that exist in current methods for complex nighttime scenes, and proposes FGDNet, a single-step diffusion framework based on Laplacian pyramid frequency priors for nighttime flare removal. This method integrates stable diffusion models with frequency prior guidance to achieve high-fidelity flare removal without relying on flare annotations. First, a Latent Diffusion Deflare Module (LDDM) is proposed, which performs flare removal and preliminary background reconstruction through single-step diffusion with LoRA fine-tuning. Simultaneously, the proposed Multi-scale Frequency Injection Module (MFIM) utilizes Laplacian pyramid decomposition to extract high-frequency details, aligns realistic textures, and injects them into the VAE decoder to restore fine structures in the preliminary flare removal results. Subsequently, the proposed Multi-band Frequency Fusion Module (MFFM) adaptively fuses high and low-frequency information from the preliminary deflare results and input images through a multi-reference attention mechanism, further enhancing the quality of structure and color restoration. This method not only addresses the problems existing in current approaches but also provides new insights for research on flare removal while preserving background structures based on latent diffusion models.

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

# A APPENDIX

## A.1 EXPERIMENTAL RESULTS OF SYNTHETIC TEST DATASETS

We evaluated our method on the synthetic test dataset provided by (Dai et al., 2024). This test dataset contains 100 pairs of nighttime flare images (With GT). Ours1 indicates that FGDNet was trained on Flare7K training datasets, and ours indicates that FGDNet was trained on the Flare7K++ training datasets. As shown in Fig. 7 and Table 4, our method outperforms other methods in flare removal performance.

Table 4: Quantitative comparison on the Flare7K++ synthetic test dataset. ↓/↑ denote lower/higher is better. Best and second-best results are in black and underlined, respectively.

| Metric\Method | Previous work | Works trained on Flare7K | | | Works trained on Flare7K++ | | | |
| | Wu | Dai | Zhou | Ours1 | Dai++ | Kotp | Ma | Ours |
|---|---|---|---|---|---|---|---|---|
| PSNR↑ | 28.26 | 30.13 | 28.78 | **30.17** | 29.50 | 29.57 | 29.58 | **29.75** |
| SSIM↑ | 0.954 | **0.965** | 0.939 | 0.963 | 0.962 | 0.961 | **0.966** | 0.965 |
| LPIPS↓ | 0.0331 | 0.0205 | 0.0286 | **0.0200** | 0.0428 | 0.0205 | 0.0200 | **0.0197** |

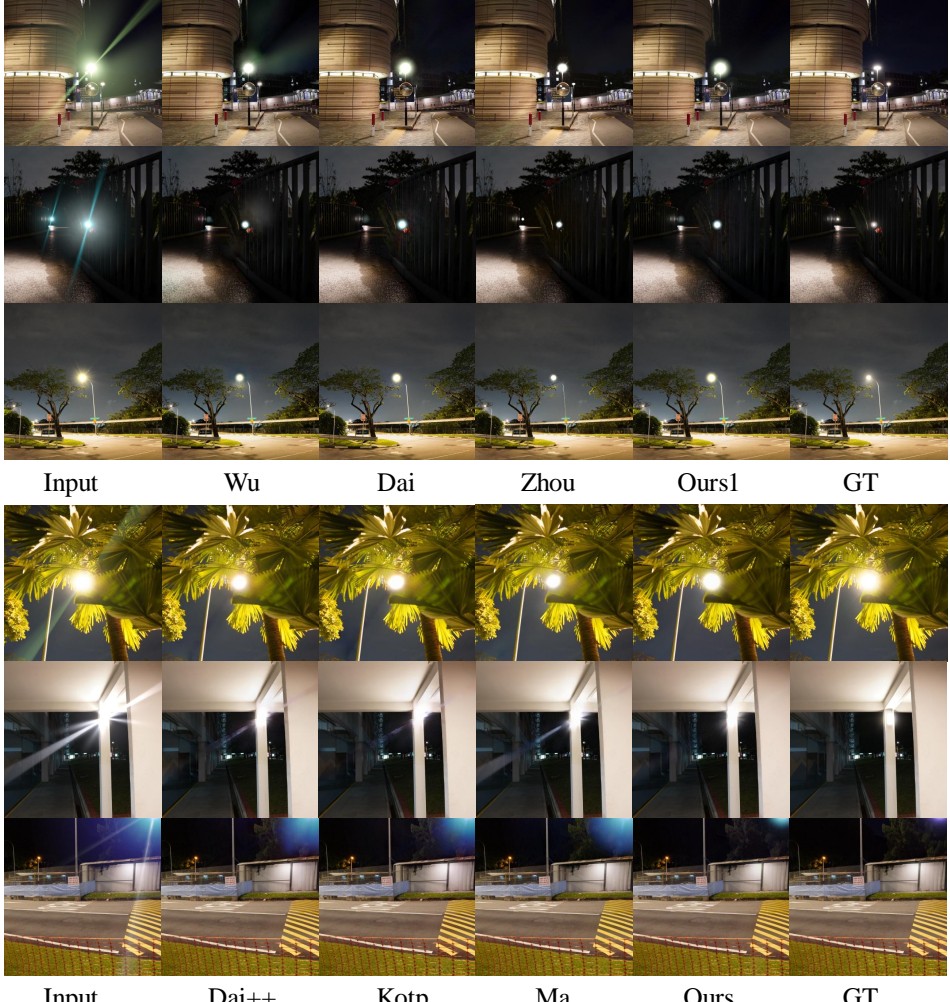

Figure 7: Qualitative comparison on the synthetic paired test dataset (with GT) provided by (Dai et al., 2024).

## A.2 MORE VISUAL EFFECTS

We provided additional visual results on the real flare-damaged image test datasets provided by (Dai et al., 2024) and (Zhou et al., 2023). As shown in Fig. 8 and Fig. 9, our method demonstrates superior visual performance on real-world night-time flare-damaged images compared to other approaches. Concurrently, we tested flare-damaged images captured using various consumer electronic devices, as illustrated in Fig. 10. Our method exhibits favourable results in both flare removal and background preservation.

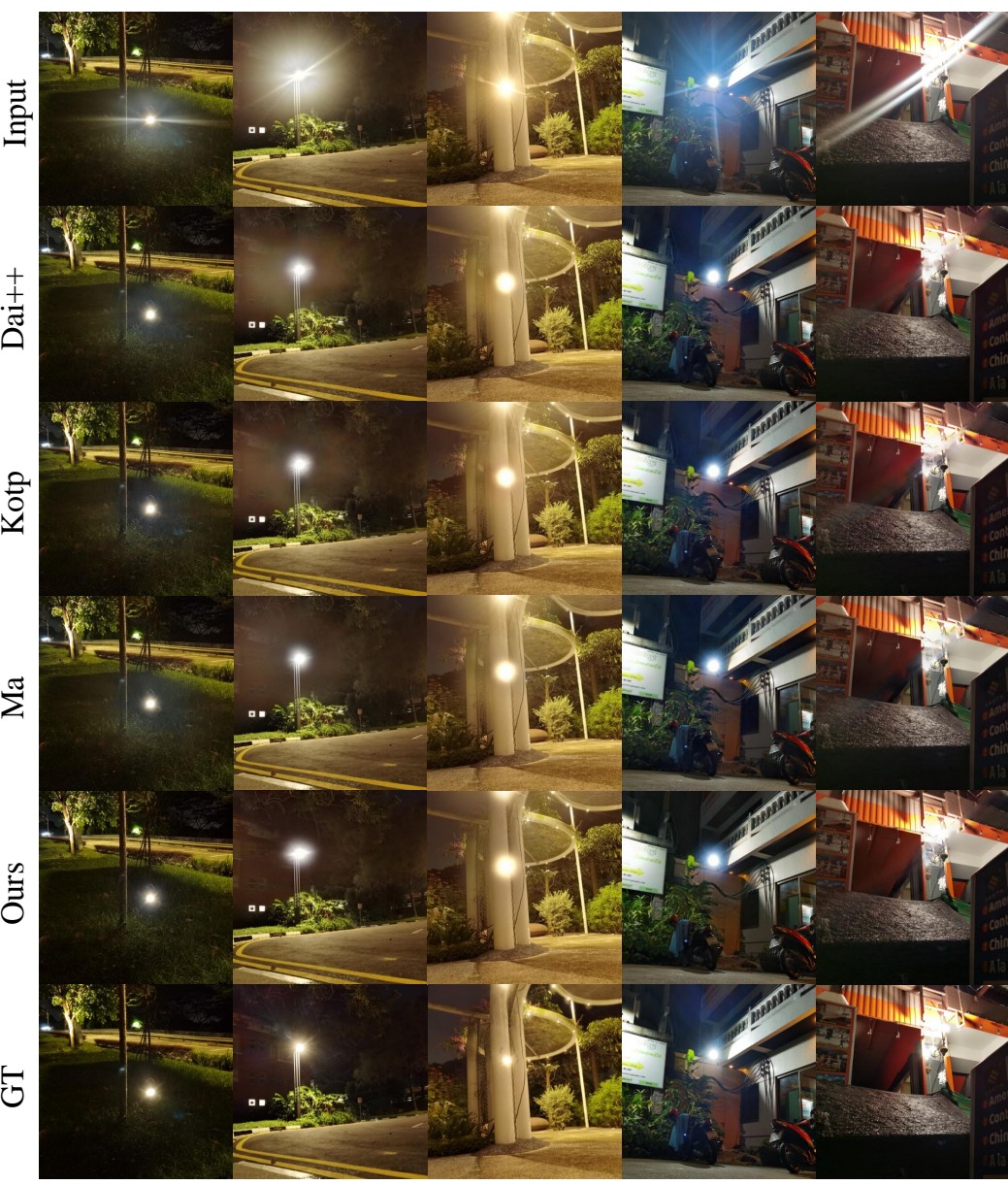

Figure 8: Qualitative comparison on the real-world paired test dataset (with GT) provided by (Dai et al., 2024). Ours indicates that FGDNet was trained on the Flare7K++ training dataset.

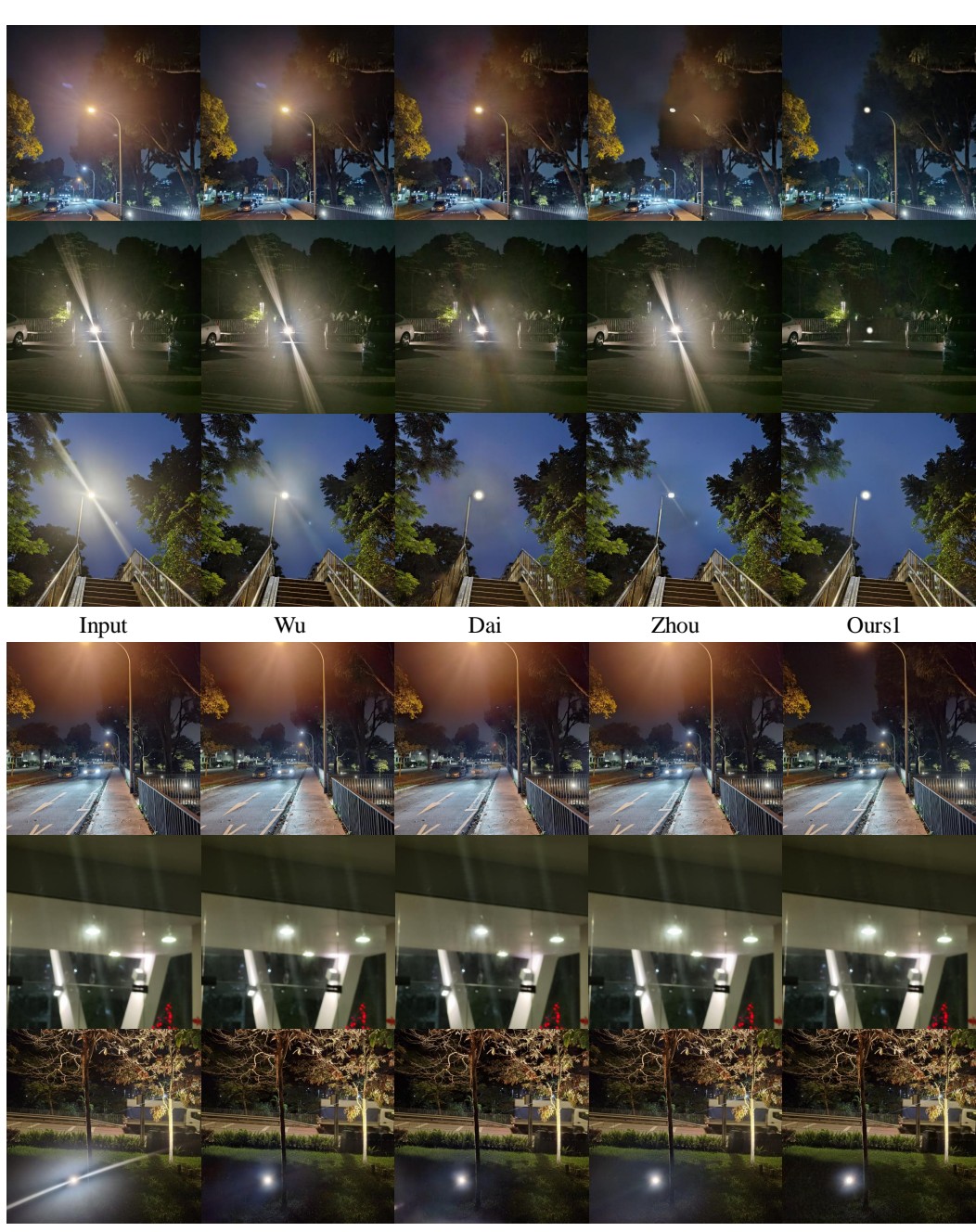

Figure 9: Qualitative comparison on the real-world captured dataset (without GT) provided by (Dai et al., 2024). Ours1 indicates that FGDNet was trained on Flare7K training dataset, and ours indicates that FGDNet was trained on Flare7K++ training dataset.

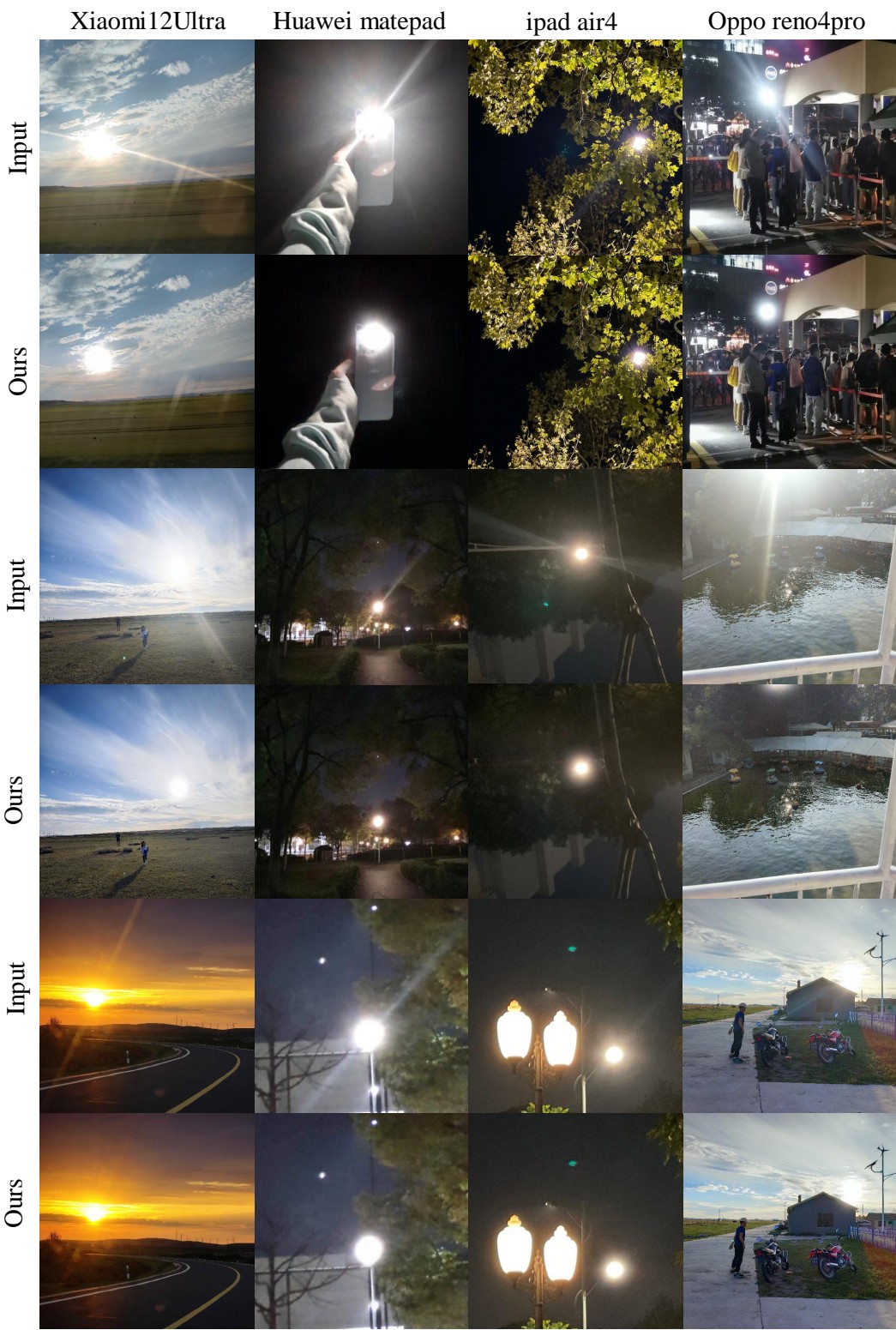

Figure 10: Qualitative comparison on the real-world captured dataset (without GT) by diverse consumer electronics (The dataset is provided by (Zhou et al., 2023)). Ours indicates that FGDNet was trained on the Flare7K++ training dataset

### A.3 USE OF LLMS

We utilized Large Language Models (LLMs), including ChatGPT and related systems, during the preparation of this paper. Their use was strictly confined to assisting with language-related tasks, such as grammar correction, spelling checks, and wording refinement, to enhance the manuscript's clarity and readability. All scientific ideas, experimental design, analysis, and conclusions were independently formulated and verified by the authors.

