# OpenReview forum: "Frequency-Prior Guided Diffusion for Nighttime Flare Removal"
_ICLR.cc/2026/Conference — ICLR 2026 Conference Withdrawn Submission_

### Official Review · Reviewer_CNxv · 2025-10-19

**Soundness:** 3
**Presentation:** 2
**Contribution:** 3
**Rating:** 4
**Confidence:** 4

**Summary:**

The paper proposes FGDNet, a frequency-guided single-step diffusion framework for nighttime flare removal. It builds on Stable Diffusion and introduces Laplace Pyramid frequency priors to enhance texture fidelity and color consistency. FGDNet integrates three main modules: (1) Latent Diffusion-based Deflare Module (LDDM): Performs single-step diffusion in latent space with LoRA fine-tuning, generating a preliminary deflared image while maintaining structural fidelity. (2) Multi-scale Frequency Injection Module (MFIM): Decomposes flare-degraded images using a Laplacian pyramid, aligns high-frequency textures, and injects them into the VAE decoder to recover fine details. (3) Multi-band Frequency Fusion Module (MFFM): Uses multi-reference attention to adaptively fuse high- and low-frequency components from the input and preliminary output, improving color and structure restoration.

**Strengths:**

1. The combination of Laplacian pyramid decomposition with a single-step latent diffusion model is novel in nighttime flare removal. The approach effectively leverages frequency priors to control generative behavior, avoiding the structural distortion and color shift problems common in multi-step diffusion.
2. This paper is well structured, well motivated and supported by experimental results.

**Weaknesses:**

**Major Weaknesses:**
1. While the combination of frequency priors and single-step diffusion is interesting, each component individually (Laplace pyramid decomposition, LoRA fine-tuning, and single-step diffusion) has been explored in prior literature. The novelty lies mainly in the integration and application, not in new theoretical contributions.
2. The paper claims efficiency through single-step diffusion, but actual runtime or FLOPs are not reported, including multi-step diffusion models.
3. Lines 100–107, The stated *contributions* largely duplicate the descriptions of the *key components*. To avoid redundancy and improve conciseness, the contribution section could be removed.
4. Line 194, the nature of the text prompt extracted by the prompt encoder is unclear. The authors should clarify what type of prompt is generated, how it differs from manually defined prompts used in standard text-to-image diffusion models, and how the system’s performance changes when no prompt is employed.
5. Line 198, The statement “However, these methods suffer from texture distortion due to iterative noise” lacks empirical evidence. The paper would be strengthened by including visual comparisons illustrating the texture distortion in conventional multi-step diffusion models versus the proposed FGDNet.
6. Although the related work section cites FF-Former, the quantitative comparison in Table 1 omits this method. Moreover, existing results in the literature suggest that FF-Former achieves competitive or even superior performance with lower computational complexity. Given that FF-Former (CVPRW 2023) remains a strong baseline, the absence of comparison and the potential higher cost of FGDNet raise concerns about the claimed advancement and efficiency.
7. In Figure 6(a), the color shift in the LDDM+MFIM output appears more severe than in LDDM alone. The authors should explain why the inclusion of MFIM, which aims to restore texture and structure, might lead to increased chromatic distortion.
8. FlareReal600, a real-world 4K dataset, is not used in the evaluation. This omission limits the evidence of generalization and robustness to high-resolution real data.

**Minor Weaknesses:**
1. Line 171, the subscripts in h(d)1 and h(d)2 appear to be incorrect.
2. Lines 437 and 463, Table3 should be Table2?

**Questions:**

See weaknesses.

If the authors can adequately address the above concerns outlined in the weaknesses section, I would be willing to reconsider my overall assessment.

---

### Official Review · Reviewer_Up5t · 2025-10-31

**Soundness:** 2
**Presentation:** 2
**Contribution:** 2
**Rating:** 4
**Confidence:** 3

**Summary:**

The paper proposes FGDNet, a nighttime flare removal framework that combines one-step latent diffusion with Laplacian-pyramid frequency priors. It comprises LDDM (one-step diffusion + LoRA), MFIM (multi-scale high-frequency alignment injected into the VAE decoder), and MFFM (multi-band fusion with multi-reference attention), trained end-to-end with pixel/perceptual and frequency losses. On Flare7K/Flare7K++ it reports modest gains on PSNR/SSIM/LPIPS and non-reference metrics (MUSIQ, MANIQA), with ablations showing incremental benefits from MFIM/MFFM.

**Strengths:**

1.Clear motivation: addresses residual artifacts/color shifts in encoder–decoder baselines and error accumulation in multi-step diffusion with a faster one-step alternative plus frequency guidance.
2.Cohesive design: frequency semantics (HF→detail, LF→illumination/color) are explicitly wired into the reconstruction path and subsequent fusion, yielding an intuitive pipeline.
3.Practicality: parameter-efficient LoRA, single-step inference, and no reliance on flare annotations while staying competitive with label-dependent methods.

**Weaknesses:**

(1) The paper claims a strong step–quality–latency advantage from the proposed one-step variant, but this trade-off is not actually quantified. There is no systematic comparison of reconstruction quality (e.g., PSNR / LPIPS / MUSIQ) versus the number of generation steps (1/2/4/8), under equalized compute and matched schedulers. Without a curve and statistical tests, it is unclear whether one-step is truly optimal or just a particular operating point.
(2) The MFIM module is trained with GT-aligned guidance, but the paper does not convincingly show how robust this alignment strategy is in deployment, where perfect GT alignment is not available and domain shift (camera, lighting, spectral response, etc.) is expected. There is no cross-camera / cross-city / cross-lighting breakdown, no analysis of variance across these splits, and no sensitivity analysis to hyperparameters like LP depth or frequency weighting. This makes it hard to assess how well the method generalizes outside the lab setting.
(3) Reported improvements may not be statistically reliable. Several gains are small (on the order of 0.1–0.6 dB PSNR), and in at least one case the method is not the best on all metrics (e.g., LPIPS). However, the paper does not report mean ± std across multiple seeds, paired significance tests, confidence intervals, or effect sizes. So it is unclear whether the improvements are consistent or just within noise.
(4) The fairness of the comparisons and the efficiency claim are under-supported. The paper does not clearly control for training budget, data augmentation strength, parameter counts, backbone pretraining, post-processing, or inference resources. There is also no summary table reporting params / FLOPs / latency / VRAM side-by-side with quality metrics. Without strict accounting, it is difficult to tell if the reported efficiency comes from algorithmic gains or simply from using more compute or heavier restoration steps than the baselines.
(5) The paper lacks qualitative/quantitative analysis of failure modes and extreme cases. There is no examination of when the model breaks (e.g., severe lighting, unusual sensors, heavy misalignment), how errors manifest, or how bad the tails of the distribution look. This limits our understanding of reliability and practical risk.

**Questions:**

1.Where is the step–quality–latency trade-off?
You claim one-step reduces accumulated error, but there’s no systematic curve vs. multi-step latent diffusion under equal compute and matched schedulers. Provide PSNR/LPIPS/MUSIQ vs. steps (1/2/4/8), latency/FLOPs, and significance tests.
2.GT-aligned MFIM: how robust is it without GT at deployment?
MFIM learns with GT alignment; in the wild there’s domain shift (devices/lenses/spectra). Add cross-camera/city/lighting splits, report variance, and include sensitivity to LP depth n and frequency-layer weights; show failure when alignment is imperfect.
3.Are the improvements statistically reliable and consistent?
Report mean±std across multiple seeds and paired tests. Some gains are small (e.g., ~0.1–0.6 PSNR), and at least one metric (e.g., LPIPS) is not SOTA in the strongest setting—quantify confidence intervals and effect sizes.
4.Fairness and resource control in comparisons?
Exactly match training budget, augmentations, parameter counts, pretraining backbones, and post-processing (e.g., light-source restoration) across methods. Provide a table with params/FLOPs/inference time/VRAM alongside metrics to justify the efficiency claim.
5.Failure modes and extreme cases?
Analyze cases with rainbow ghosts, streaks, large reflections, heavy backlight, fog/neon, etc. Include ΔE color error and more structure-sensitive measures; visualize frequency/attention maps. Please state when one-step fails and whether multi-step fallback helps.

---

### Official Review · Reviewer_A6A4 · 2025-11-01

**Soundness:** 3
**Presentation:** 3
**Contribution:** 3
**Rating:** 6
**Confidence:** 4

**Summary:**

This paper proposed FGDNet, a single-step diffusion framework guided by Laplace Pyramid frequency priors to address residual artifacts, color shifts, and background distortion in existing nighttime flare removal methods. Experiments on Flare7K and Flare7K++ datasets show superior performance in both quantitative metrics and qualitative results. The main contributions are the novel single-step diffusion framework with frequency prior guidance.

**Strengths:**

1.	The paper achieves originality through the creative combination of existing techniques. While single-step diffusion and frequency-aware design are individually known, integrating Laplace Pyramid frequency priors into a single-step diffusion framework for nighttime flare removal is a novel exploration to some extent.

2.	The method design is theoretically coherent. Laplace Pyramid decomposition is reasonably applied, as low-frequency components dominate illumination/color and high-frequency components dominate texture/structure, aligning with the characteristics of flare degradation (affecting both frequency bands). The training strategy combines data consistency loss and frequency consistency loss, covering both pixel and frequency dimensions to ensure restoration quality.

3.	The work addresses a practical problem in nighttime photography (flare removal) and provides a new paradigm for latent diffusion-based image restoration.

**Weaknesses:**

1.	Although the combination of techniques is innovative to some extent,  but the core components lack fundamental novelty. Single-step diffusion for image restoration was previously proposed, and frequency-aware integration (e.g., Laplace Pyramid) is common in image processing. The paper does not sufficiently discuss how its frequency-guided single-step diffusion differs from prior combinations (e.g., whether other works have applied frequency priors to flare removal).

2.	The related work section fails to cover recent diffusion-based image restoration methods that integrate frequency information. For example, if there are works using frequency priors in latent diffusion for denoising or deblurring, the paper should compare and contrast with them to highlight its advantages. Additionally, the discussion on encoder-decoder vs. diffusion-based methods is superficial, lacking analysis of why frequency guidance is more effective for diffusion models in flare removal.

**Questions:**

1.	Have there been prior works applying Laplace Pyramid frequency priors to diffusion-based image restoration (especially flare removal)? If yes, how does FGDNet differ in module design and guidance mechanism?

2.	Have you tested FGDNet on flare images captured in extreme conditions (e.g., strong backlight, rainy nights) or with low-end cameras? How does it perform compared to state-of-the-art methods in these scenarios?

3.	The paper claims no need for flare annotations. Can you explain how the text prompt extractor ensures accurate guidance for flare removal, especially in complex scenes with multiple light sources?

---

### Official Review · Reviewer_28kX · 2025-11-01

**Soundness:** 2
**Presentation:** 2
**Contribution:** 2
**Rating:** 4
**Confidence:** 4

**Summary:**

This paper introduces FGDNet, a single-step diffusion model for nighttime flare removal, guided by Laplacian-pyramid frequency priors. Unlike traditional multi-step diffusion methods, which suffer from error accumulation and color distortion, FGDNet achieves stable, high-fidelity restoration without requiring flare annotations. It combines three modules: LDDM for latent flare removal using one-step diffusion with LoRA fine-tuning, MFIM for restoring fine details by aligning and injecting high-frequency components into the decoder, and MFFM for enhancing structural and color fidelity by fusing the deflared output with input frequency information through multi-reference attention.

**Strengths:**

The paper introduces a technically novel integration of single-step latent diffusion for coarse restoration, multi-scale frequency alignment for detail refinement, and attention-based frequency fusion for perceptual enhancement, forming a coherent restoration pipeline that effectively improves structural and color fidelity in nighttime flare removal.

**Weaknesses:**

- The authors emphasize that their model does not rely on flare labels, which they present as an advantage. However, this claim is not sufficiently validated. It remains unclear whether omitting flare labels improves generalization or is merely due to dataset constraints.

- Although Table 3 shows numerical improvements, the visual quality improvement is not clearly supported by the provided figures. The claimed enhancement in flare removal and color retention (L459–460) appears unconvincing based on the qualitative results.

- The color restoration claim (L459–460) appears inconsistent with the visual examples. The red-boxed regions in Fig. 6(a) still show noticeable color deviation from the GT.

- The discussion around Table 3 is somewhat confusing, as it is cited in different contexts (MFIM vs. MFFM), making it unclear which experiment corresponds to which module.

- Overall, while the quantitative gains are evident, the manuscript lacks sufficient qualitative evidence to support the claimed improvements in flare removal and color fidelity.

- Minor editorial issues (typos, grammar, and terminology/clarity):
   - L151: Missing space before proposes — “(Liang et al., 2021)proposes” should have a space.
   - L157: The term “LUT” appears without prior definition.
   - L282–L283: The clause with based on is an incomplete sentence.

**Questions:**

- In L263, MFIM is described as being designed with $n = 4$, but in Figure 2 only three high-frequency components $h_1^{(f)}$, $h_2^{(f)}$, $h_3^{(f)}$. Does $L^{(f)}$ correspond to the low-frequency input $L^{(f)}$ to the VAE encoder? Including $L^{(f)}$ explicitly in the figure would clarify the description.

- In Fig. 6(a), does the second row correspond to the color map? This should be explicitly stated in the Figure caption for clarity.

- In Fig. 6(a), none of the module combinations visually appear to match the GT color closely. Could the authors clarify whether the improvement is quantitatively or perceptually significant?

- Table 3 is referenced multiple times in the text (for both MFIM and MFFM). Does Table 3 report results for both modules, or are these separate experiments? If the latter, the authors should separately report the ablation for MFIM and MFFM with and without multi-reference attention (MA).

---

### Note · Authors · 2025-11-13

I have read and agree with the venue's withdrawal policy on behalf of myself and my co-authors.